# Azacitidine Plus Venetoclax for the Treatment of Relapsed and Newly Diagnosed Acute Myeloid Leukemia Patients

**DOI:** 10.3390/cancers14082025

**Published:** 2022-04-16

**Authors:** Sylvain Garciaz, Marie-Anne Hospital, Anne-Sophie Alary, Colombe Saillard, Yosr Hicheri, Bilal Mohty, Jérôme Rey, Evelyne D’Incan, Aude Charbonnier, Ferdinand Villetard, Valerio Maisano, Laura Lombardi, Antoine Ittel, Marie-Joelle Mozziconacci, Véronique Gelsi-Boyer, Norbert Vey

**Affiliations:** 1INSERM, CNRS, Department of Hematology, Institut Paoli-Calmettes, Aix-Marseille University, 13009 Marseille, France; 2Deparment of Hematology, Institut Paoli-Calmettes, 13009 Marseille, France; hospitalm@ipc.unicancer.fr (M.-A.H.); saillardc@ipc.unicancer.fr (C.S.); hicheriy@ipc.unicancer.fr (Y.H.); mohtyb@ipc.unicancer.fr (B.M.); reyj@ipc.unicancer.fr (J.R.); dincane@ipc.unicancer.fr (E.D.); charbonniera@ipc.unicancer.fr (A.C.); villetardf@ipc.unicancer.fr (F.V.); maisanov@ipc.unicancer.fr (V.M.); lombardil@ipc.unicancer.fr (L.L.); 3Department of Molecular Biology, Institut Paoli-Calmettes, 13009 Marseille, France; alarya@ipc.unicancer.fr; 4Department of Biopathology, Institut Paoli-Calmettes, 13009 Marseille, France; ittela@ipc.unicancer.fr (A.I.); mozziconaccimj@ipc.unicancer.fr (M.-J.M.); gelsiv@ipc.unicancer.fr (V.G.-B.); 5Predictive Laboratory Oncology, Equipe Labellisée Ligue Contre le Cancer, Centre de Recherche en Cancérologie de Marseille (CRCM), Institut Paoli-Calmettes, INSERM UMR 1068, CNRS UMR725, Aix-Marseille Université, 13007 Marseille, France

**Keywords:** acute myeloid leukemia, venetoclax

## Abstract

**Simple Summary:**

The combination of venetoclax and azacititine (VEN–AZA) has recently been approved for the treatment of unfit newly diagnosed (ND) acute myeloid leukemia (AML) patients. Few data are available for the relapsed and/or refractory (R/R) AML group. We retrospectively compared the outcome of 39 R/R to 38 concomitant ND AML patients treated in our institution between 01/20 and 12/21. Response rates were lower in R/R AML (37% versus 56%); adverse cytogenetics was associated with treatment failure only in the R/R group (Relative Risk = 0.10, *p* = 0.005). *ASXL1*, *IDH* and *SFSR2* mutations were associated with a trend in a higher response rate in the R/R group. Median leukemia-free survival was not different between the two groups (9.4 months and 10.3 months in the ND and R/R groups, respectively). In conclusion, VEN–AZA can be efficient as a salvage treatment for selected R/R AML patients.

**Abstract:**

Venetoclax (VEN) belongs the BH3-mimetic class that selectively targets BCL-2, activating apoptosis. The combination of VEN and azacitidine (AZA) has changed the paradigm of treatment of newly diagnosed (ND) acute myeloid leukemia (AML) patients ineligible for intensive chemotherapy. There is scarce evidence for the use of VEN–AZA for relapsed or refractory (R/R) AML. We compared the outcome of 39 R/R AML and 38 ND AML patients treated between 01/20 and 12/21. The median age was 69 (22–86) and 73 (61–81) in the R/R and ND groups, respectively. Adverse cytogenetics were found in 36% of patients in the R/R group and 59% of patients in the ND group. Overall response rate was 37% in R/R AML, including 13% CR, 8% CRi, 3% PR and 13% MLFS, and 58% in the ND AML, including 32% CR, 13% CRi and 13% MLFS. Adverse cytogenetics was associated with treatment failure in the R/R group (Relative Risk = 0.13, *p* = 0.005). Median overall survival (OS) was 5.9 months in the R/R group and 9.4 months in the ND group. Median OS was 2.2 months in the adverse cytogenetics group versus 8.7 months in the intermediate cytogenetics group in the R/R group (*p* = 0.02). Median leukemia-free survival was not different between the two groups (9.4 months and 10.3 months), indicating that VEN–AZA can be an efficient salvage treatment for selected R/R AML patients. In conclusion, VEN–AZA is a promising treatment for ND AML and for selected R/R AML patients.

## 1. Introduction

Acute myeloid leukemia (AML) is a heterogeneous group of severe diseases with various molecular alterations, including chromosomal aberrations or genes mutations, that mainly occur after the sixth decade [1]. Venetoclax (VEN) belongs to a novel BH3-mimetic class of small molecules that selectively targets BCL-2, activating the apoptosis effectors BAX and BAK to drive mitochondrial outer membrane permeabilization (MOMP), cytochrome c release and cell death [2,3]. Combination of VEN and the hypomethylating agent (HMA) azacitidine (AZA) or decitabine has deeply changed the paradigm of treatment of newly diagnosed (ND) AML patients who are not candidates for high-dose chemotherapy because of older age or comorbidities, a category of patients classically associated with poor outcomes [4,5,6]. In the phase 3 VIALE-A trial results, approximately 65% of VEN–AZA treated patients had a complete response (CR) or CR with incomplete recovery (CRi) and more than 1-year overall survival (OS). As a frame of reference, response rates and OS in the AZA alone control arm were 30% and 7 months, respectively [6]. Predictive factors associated with VEN response remain elusive but there is growing evidence to show that *NPM1* and *IDH* mutated AML patients have a good outcome, whereas AML patients with mutations in *TP53* or RAS pathway genes have a poor outcome [7,8,9,10,11].

Relapsed or refractory (R/R) AML represents a group of patients associated with an extremely poor outcome and no standard of care [12]. VEN–AZA treatment in this setting has been consistently associated with lower response and survival rates in this group than in ND patient cohorts. Approximately 20–40% of the patients have a response, with a significant part of them experiencing relatively long-term survival [13,14,15,16,17,18,19,20,21,22,23]. On the other hand, VEN–AZA is often associated with hematological toxicities, and many patients will experience febrile neutropenia and long-term hospitalization, affecting their quality of life [24,25]. Given that the drug is currently used in France “off-label” in the R/R setting, it is important to avoid a useless treatment with frequent toxicities for patients having a poor chance of response. As the predicting factors of clinical response to VEN–AZA in the relapse setting are not known, identifying the category of patients that would benefit from the treatment is a challenge for physicians.

In this study, we report a cohort of 77 patients treated with VEN–AZA in our institution, 38 upfront and 39 in the R/R setting. The objectives of the study are to study the response and survival rates in a real-life population of ND and R/R VEN–AZA-treated patients, and to study clinical and molecular characteristics associated with clinical response and outcomes.

## 2. Materials and Methods

### 2.1. Patients

We retrospectively collected clinical, biological, and molecular data from the first 83 patients treated with VEN–AZA at the Institut Paoli-Calmettes between January 2020 and December 2021. We excluded from our study patients who received VEN–AZA for treatment of molecular relapse (*n* = 4), extramedullary disease (*n* = 1) and as a consolidation treatment after intensive chemotherapy (*n* = 1). All the included patients (*n* = 77) received at least one cycle of VIDAZA and 3 days of VEN, and had more than 2 months’ follow-up. Informed consent was obtained from all subjects involved in the study following institutional guidelines, and in accordance with the Declaration of Helsinki. The study has been approved by the Institutional Review Board (VIDAZA VENETO-IPC 2020-043, approval date: 21 July 2020).

### 2.2. Patient Samples Molecular Characterization

Karyotyping and Fluorescence in situ hybridization (FISH) were performed on bone marrow or peripheral blood using standard techniques. Chromosome abnormalities were identified with RHG-banding and described according to the International Standing Committee on Human Cytogenetic Nomenclature (ISCN 2020). Molecular analyses were performed on DNA samples extracted from the bone marrow (BM). BM mononuclear cells were purified on Ficoll gradient and processed for DNA extraction using Qiasymphony DNA kit (Qiagen) on QIASYMPHONY (Qiagen) according to the manufacturer’s procedures. Molecular assessment of *NPM1*, *FLT3* was performed as previously described [26]. *JAK2*, *TP53*, *IDH1/2* status was determined either by individual gene sequencing (quantitative PCR with ipsogen^®^ Jak2 MutaScreen kit [Qiagen, Hilden, Germany]; Sanger with in-house designed protocol and Droplet Digital PCR [ddPCR] using ddPCR™ *IDH1* [R132C/L/S/G/H] and *IDH2* [R140L/W/G/Q and R172K] probes [Bio-Rad, Pleasanton, CA, USA] on the QX-200 droplet reader [BioRad] or by Next-Generation sequencing [NGS]). Mutations in a custom targeted panel of 60 genes (130 kpb) recurrently mutated in myeloid neoplasms (Appendix A) were screened by NGS assay using a Custom Myeloid Lymphoid Solution (SOPHIA GENETICS, Saint Sulpice, Switzerland). DNA libraries, built with capture-based enrichment protocol (SOPHIA DNA Library Prep kit, SOPHIA GENETICS, Switzerland), were sequenced using NextSeq550Dx Instrument (Illumina, San Diego, CA, USA). Data analyses were performed using 2 commercial bioinformatics pipelines (Sophia Genetics DDM^®^ and CLC Genomics Workbench, Biomedical Genomics Analysis—BGW software/QIAGEN). Interpretation used public databases (gnomAD, COSMIC, dbSNP, ClinVar) for variant annotations and predictive in silico tools (SIFT, PolyPhen-2, CADD, MutationTaster) in case of unknown variant. The sensitivity of the technique was about 2 %, depending on the depth quality of the average of the specific coverage of each locus and each sample. Risk group categories were assigned according to the 2017 ELN risk stratification [27].

### 2.3. Treatment Modalities

Patients received AZA at standard dose of 75 mg/m^2^ QD for seven days, and VEN was administrated either at 400 mg or 100 mg when associated with strong Cytochrome P450, family 3, subfamily A (CYP3A) inhibitors after three days’ ramp up (Figure 1). The first cycle was administered in the in-patient unit of hematology. During the first cycle, VEN was given for 14 to 28 days, depending on age and comorbidities. The second cycle was started on day-28, when possible, and patients received 7 to 28 days of VEN depending on bone-marrow evaluation and hematological toxicities. For responding patients receiving subsequent cycles, the higher dose with minimal toxicities was achieved with G-CSF utilization as recommended. Bone marrow assessment was performed during the first cycle between day-21 and day-35, and subsequently based on physician discretion.

### 2.4. Assessment of Response

Response to VEN–AZA was determined using the ELN 2017 criteria [27]. The ORR was defined as the combination of complete response (CR), CR with incomplete hematologic recovery (CRi), and morphologic leukemia-free state (MLFS).

### 2.5. Statistical Analysis

Patient characteristics were summarized using median (range) for continuous variables and frequency (percentage) for categorical variables. Categorical variables were compared for significance using the Fisher’s exact test, and continuous variables were analyzed using the Student *t*-test. Relative risk measured the probability of event in exposed group/ probability of event in not exposed group [28]. Statistical analyses were conducted with PRISM 5.0 SPSS statistics 22. Logistic and COX regressions were performed as previously described [29]. Time to progression was measured as the interval between the start of treatment and relapse after censoring death before relapse and lack of response. Overall survival (OS) was measured as the time from VEN–AZA initiation to date of death or date of last follow-up (censored). Event-free survival (EFS) was measured from VEN–AZA initiation to date of death, progression, whichever came first, or date of last follow-up (censored). Leukemia-free survival (LFS) was measured as the time from the date of remission (including CR, CRi or MLFS) to the time of relapse, death, or date of last follow-up (censored) [27]. All survival endpoints were calculated by the Kaplan–Meier method using the log-rank test [30]. Significance was defined as a *p* value of <0.05.

## 3. Results

### 3.1. Patient Clinical and Molecular Characteristics

We retrospectively included 77 patients treated with VEN–AZA during the period. Thirty-eight patients received VEN–AZA upfront for ND AML, in accordance with the European Medicines Agency authorization. The other 39 patients received VEN–AZA for R/R AML. Median age was 72 (73 in the first line group and 69 in the R/R group, *p* < 001). In all, 76% of the patients in the first line group and 49% in the R/R group had a secondary AML (*p* = 0.035), including myelodysplastic-related changes (MRC), post myeloproliferative neoplasm (MPN) and therapy-related (TR) AMLs. In the ND cohort, 16% of the patients received prior AZA treatment for a history of MDS. In the R/R group, 26% had received AZA and 79% prior chemotherapy. Nine patients (12%) had a *FLT3* mutation (7 in the R/R group and 2 in the ND group). All the R/R patients received FLT3 inhibitor (midaustorin or gilteritinib) in combination with upfront chemotherapy or at first relapse. Ten patients had relapsed after a prior HSCT. In all, 59% of the patients in the ND cohort and 39% in the R/R had adverse cytogenetics (*p* = 0.069). *NPM1* and *FLT3* mutations were found in 8% and 5% in the ND group and 18% and 18% in the R/R group, respectively. *IDH* and *TP53* mutation status were available for 75 and 67 patients, respectively. NGS was available for 54 patients (30 in the ND group and 24 in the R/R group). Clinical and biological characteristics of the 77 patients treated with VEN–AZA are summarized in Table 1. The whole set of molecular alterations are represented in Appendix A. Among the main mutated genes in the first-line group, we noted *ASXL1*, *RUNX1* and *SRSF2*, reported in 33%, 33% and 24%, respectively, consistently with a strong enrichment in secondary AMLs in this cohort (Figure 2).

### 3.2. Treatment

All the patients received a first cycle of VEN–AZA, and most of them were given 100 mg VEN (67, 87%) in combination with azoles administered as antifungal prophylaxis. The median time of VEN was 21 days (range = 3–28). Four patients (three from the R/R group and one from the ND group) received less than 14 days of VEN during the first cycle, three because of rapid progression and one because of poor general status. The 73 other patients received more than 14 days of VEN. Fifty-four patients out of the seventy-seven (71%) received a second course of VEN–AZA. Reasons for VEN–AZA discontinuation after cycle 1 for 23 patients were death (9 patients, 11.6%), progression (7 patients, 9.1%), toxicities (6 patients, 7.7%) or loss of follow up (1 patient, 1.3%). Median time between the first and second cycles was 34 days (range 22–93). Median number of VEN–AZA cycles was two (range 1–12). Median duration of hospitalization for cycle 1 was 30 days (ranges 1–60). Serious adverse events during the two first cycles were mainly febrile neutropenia and grade 3/4 hematological toxicities consistent with previous reports [7].

### 3.3. Treatment Response

Response rate assessed on day-28 and day-56 are shown in Figure 3A–C and Appendix A. In total, 58% of the patients experienced a response, including CR, PR, CRi and MLFS in the ND AML group, and 37% in the R/R cohort. After excluding post-MPN, in accordance with the VIALE-A study inclusion criteria [6], ORR was 66% in the ND AML group patients and 40% in the R/R group (Figure 2 and Appendix A). Response rate was poorer in adverse cytogenetics in R/R AML patients. Amongst the AML samples with NGS data available (*n* = 54), we pooled the 14 *RAS* mutated AML with the 10 mutated *TP53* mutated AML to obtain a group of 22 patients (two AML patients had both RAS and *TP53* mutations); in this group of AML patients with *TP53* and/or *RAS* mutated, response rate was lower (31% vs. 66%, Figure 3D–E and Appendix A). Finally, we aimed to study the impact of the first cycle VEN dosage and duration of treatment on response rate. We considered that 100% of the initial VEN dose was achieved when patients received one complete cycle of 28 days of VEN 400 mg (or 100 mg if associated with azoles) as previously published in the VIALE-A study [6]. We excluded patients with severe renal failure (*n* = 4) from this analysis. In total, 51 patients received more than 50% of the total dose, and 22, 50% or less than the total dose. The response rate was not different between these two groups (Figure 3F and Appendix A).

### 3.4. Survival Analyses

Median OS and EFS were 9.4 months and 5.8 months in the first line cohort and 5.9 and 2.3 months in the R/R cohort (Appendix A), respectively. As anticipated by response rates, OS was significantly lower in the R/R group with adverse cytogenetics (*p* = 0.02) and the *TP53* and/or *RAS* mutated patients (*p* = 0.009, Figure 4A–D). We did not observe any influence of the VEN dose during the first cycle on survival (Appendix A). We registered 12 deaths (15%) during the first 56 days of treatment (6 in the ND and 6 in the R/R cohort). Thirteen deaths were noted during the first six months after treatment initiation in the ND cohort. Two patients died from non-relapse mortality (sepsis). The other patients relapsed rapidly after VEN–AZA initiation and died because of the progression of the disease. Most of these patients had poor-risk cytogenetics status (9/13) or *TP53* and/or *RAS* mutations (7/13).

When considering only responding patients, the median LFS was 9.4 months and 10.3 months in the ND and R/R groups, respectively (*p* = 0.78, Figure 4E), indicating that once achieved, responses can be sustainable in the R/R AML. MLFS status on day-56 was not associated with a worse outcome than CR/CRi/PR status both in OS and LFS analyses (Figure 4F). Duration of response was 203 days (94–400). Among the responding patients, 9 received HSCT (including 4 in the R/R group), and 3 of them relapsed between 83 and 231 days after transplantation.

### 3.5. Risk Factor Associated with Response and Survival

We observed that disease status and adverse cytogenetics were associated with a trend to a lower response rate and poorer survival in multivariate analyses in the whole cohort of VEN–AZA-treated patients (*n* = 77, Table 2).

We next studied the clinical, biological or molecular factors associated with response within the two groups of VEN–AZA-treated AML patients (ND versus R/R). We compared the responding patients on day-56, called thereafter «responders» and the patients who died or did not respond on day-56 («non-responders»). We excluded two non-evaluable patients from the ND AML group because of loss of follow-up. We focused our analysis on the R/R patient group as it was not clear which patient may benefit or not from VEN–AZA. As a frame of reference, we performed the same analysis in the ND AML group of patients. Although no clear clinical, biological or molecular data were associated with response in our ND-AML cohort, strikingly, adverse cytogenetics such as complex and monosomal karyotypes were clearly associated with a lack of response in the R/R group (Relative risk of response = 0.1 [0.02–0.7], *p* = 0.005, Figure 5 and Appendix A). We also observed a trend in a higher response rate in the *IDH*, *ASXL1* or *SFRS2* mutated group of R/R AML patients, but this did not reach statistical significance. The presence of *TP53* and/or *RAS* mutations were associated with a trend toward a lower response rate in the R/R group (Figure 5).

## 4. Discussion

We performed a retrospective analysis of VEN–AZA-treated AML patients in a single institution between 01/20 and 12/21. We found that R/R AML had 37% ORR and median OS was 5.9 months, whereas ND patients had 58% ORR and 9.4 months median OS. Consistently with other studies, we identified that R/R AML patients have poorer outcomes in multivariate analyses than ND AML patients. We observed that adverse cytogenetics was a predictor of poor response mainly in the group of R/R AML. Although it is widely admitted that VEN–AZA has changed the outcome of non-previously treated elderly or unfit AML patients, it is less clear whether this treatment may be beneficial for R/R AML patients. The only prospective studies using VEN in the setting of R/R AML are two phase II studies [31,32]. In the seminal study, 800 mg VEN was given in monotherapy to 30 R/R patients and 2 high-risk ND AML patients. Response rate was 19%, and 19% of the patients had a bone marrow blast count reduction. *IDH* mutation was associated with a higher response rate (33%) [31]. The second study enrolled 168 patients, including 55 R/R AML patients treated with 400 mg VEN combined with 10-day decitabine. In this group, median age was 62 years and 64% had ELN adverse-risk disease, including 42% with adverse cytogenetics. Overall response rate, including CR and CRi, was 34% and 10% had MLFS. The median OS in the R/R AML patients was 7.8 months, and the median duration of response was 16.8 months. Higher response rates and longer survivals were observed in the *NPM1* and *IDH*-mutated groups of patients whereas *TP53*-mutated patients had worse outcomes [32].

Our results are consistent with the retrospective/real life studies that specifically assessed VEN–HMA treatment in R/R AML [14,16,17,19]. In a recent monocentric study from the Mayo Clinic reporting a cohort of 42 R/R AML patients treated with VEN–HMA, the response rate was 33.3% (including 19% CR and 14.3% CRi) and a 5-month OS [14]. Another retrospective monocentric study of 86 patients from the MSKCC included 35 treated with VEN–AZA, 20 with VEN–DEC, and 27 with VEN–ARAC. VEN–AZA patients showed a 49% ORR (CR/CRi/MLFS) and 16% PR. Median duration of response was 10.2 months and median OS was 25 months. VEN–AZA regimen was associated with better response and survival rates as compared with another VEN-based regimen but the adverse cytogenetics group was underrepresented (26%) [16].

In our study, we noted an extremely poor response rate in the adverse cytogenetics group of patients. This result may lead physicians to not propose VEN–AZA for R/R patients with poor cytogenetics. Nevertheless, the impact of cytogenetics on response or survival varied between the studies. Our results are consistent with the study from Stahl et al., showing that adverse cytogenetics, alongside *TP53* mutation, is predictive of lower odds of response and OS [16]. In the Mayo Clinic report, abnormal cytogenetics did not predict response as 57% of the responders had abnormal cytogenetics, but the detail on karyotypes is not available. Adverse cytogenetics did not predict response or survival either in another monocentric study of R/R AML patients [23].

Independently of karyotype, molecular factors driving good response or survival to VEN–AZA or VEN–DEC were *NPM1* and *IDH1* mutation, whereas *TP53*, *NRAS/KRAS*, *SF3B1*, *ASXL1* and *EZH2* were associated with poor outcome in the MSKCC study [16]. Other groups showed that *ASXL1* and *RUNX1*, two genes associated with poor response to chemotherapy, are no longer predictive in the setting of VEN–AZA [19,33]. Moreover, preclinical data indicate that RAS pathway alterations can drive VEN resistance [34] In our series, we found a trend in a lower response rate in patients with *TP53* and/or *RAS* pathway mutations and higher response rate of *ASXL1* mutation alongside *IDH* and *SFRS2* mutations. The role of *ASXL1* in response to AZA alone or in combination with VEN has been previously suggested and is probably related to a specific epigenetic profile [35,36].

Overall, it seems that molecular factors are not the only factors predicting survival. Impact of VEN dose during the first cycle was not associated with response rate in the R/R or in the ND cohort, contrary to a recent study suggesting that lower exposition to VEN may be associated with treatment failure [37]. On the contrary, prior chemotherapy exposition in some cases may have selected leukemic clones resistant to BCL2-inhibition. BH3 profiling is a functional tool which studies dependency of cancer cell to antiapoptotic protein that can be used in clinics to predict response to VEN [38,39,40]. Some of these studies have shown that BCL2 resistant cells may be primed for MCL1 inhibition [38,41]. Clinical trials using MCL1 inhibitors are currently recruiting. The effort to increase the understanding of VEN and/or AZA resistance is important as reported in recent reviews [7,11]. We have pre-clinically studied new therapeutic strategies that could induce non-canonical apoptosis independently of BCL2 in patient samples resistant to VEN [42]. Further studies are needed to confirm whether another BH3 mimetic or a new drug inducing alternative cell death could be useful in the setting of R/R AML.

Finally, about 30% R/R patients in our cohort, consistently with other published reports, experienced sustained responses to VEN–AZA leading to a relatively prolonged LFS (10.3 months). Promising results have been observed in heavily pretreated patients as those relapsing after allogenic SCT. A recent study suggested that VEN–AZA plus Donor Lymphocyte Infusions may be an option for patients relapsing after Allogenic SCT [43]. Finally, VEN–AZA can be considered as a bridge-to-transplant strategy in some selected cases. In our study, nine patients underwent allo SCT after VEN–AZA treatment, including four in the R/R group. A retrospective study published recently described a cohort of 21 patients > 60 years treated upfront with VEN–AZA, followed by allogenic SCT with good outcomes compared to patients treated with maintenance [44]. Further studies are needed to confirm the feasibility and efficacy of such a strategy for high-risk AML patients.

## 5. Conclusions

Our results showed a catastrophic response rate in R/R AML with adverse cytogenetics, but a third of R/R patients will achieve a sustainable response. Multicentric studies with a higher number of patients are needed to determine more precisely which subgroup of R/R AML will take advantage of VEN–AZA.

## Figures and Tables

**Figure 1 cancers-14-02025-f001:**
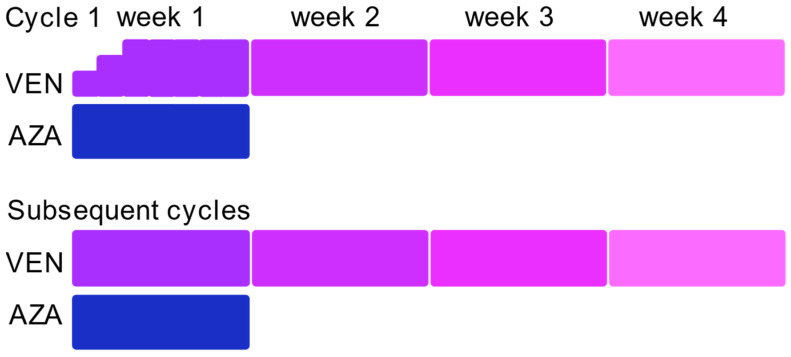
Treatment modalities. Patients received 7 days of AZA (blue). Number of VEN weeks of treatment (different shades of purple) during cycle 1 and subsequent cycles depends on patients age and comorbidities, disease response, and hematological toxicities.

**Figure 2 cancers-14-02025-f002:**
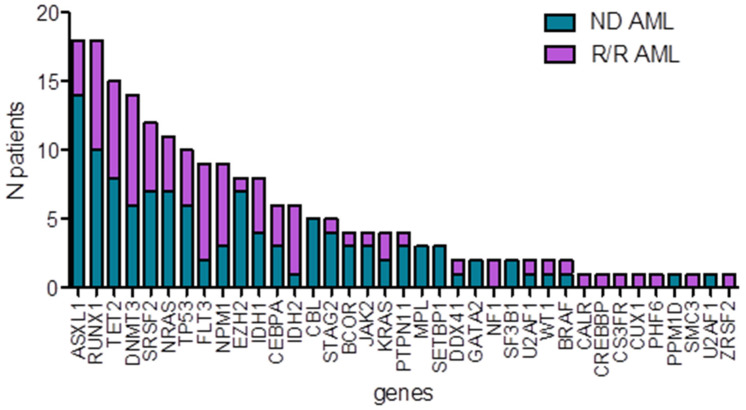
Bar graph showing the number of patients with detected mutations in the ND and R/R groups of patients treated with VEN–AZA.

**Figure 3 cancers-14-02025-f003:**
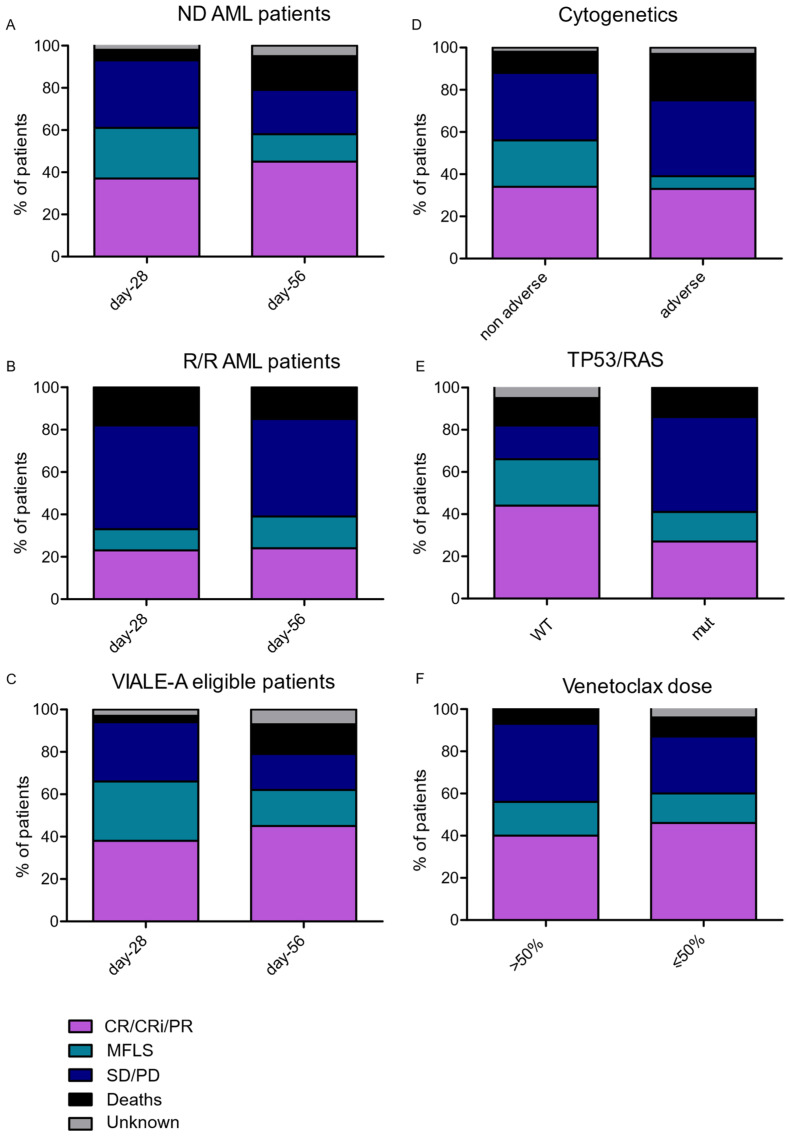
Bar graphs showing response rates on day-28 and day-56 in the first line cohort (**A**), the R/R cohort (**B**) and the VIALE-A trial eligible patients (**C**). Bar graphs showing response rate on day-56 according to cytogenetics (**D**), *TP53* and/or *RAS* mutation (**E**) and % of VEN dose at cycle 1 (**F**).

**Figure 4 cancers-14-02025-f004:**
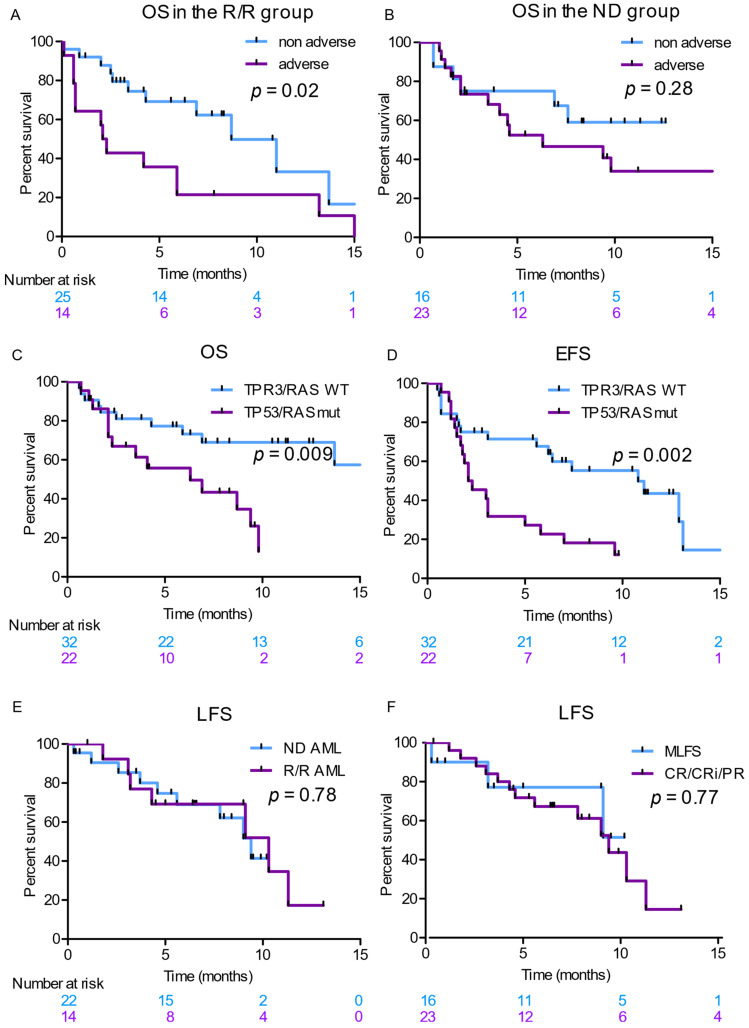
Kaplan–Meier analyses showing overall survival (OS) in the R/R AML group (**A**) and in the ND AML group (**B**) of patients according to cytogenetics, OS and event-free survival (EFS) according to *TP53* and/or *RAS* status (**C**,**D**) and leukemia-free survival (LFS) of responding patients in the R/R group ND AML group according to the disease status before starting treatment (ND versus R/R, (**E**) or response on day-56 (MLFS versus CR/CRi/PR, (**F**).

**Figure 5 cancers-14-02025-f005:**
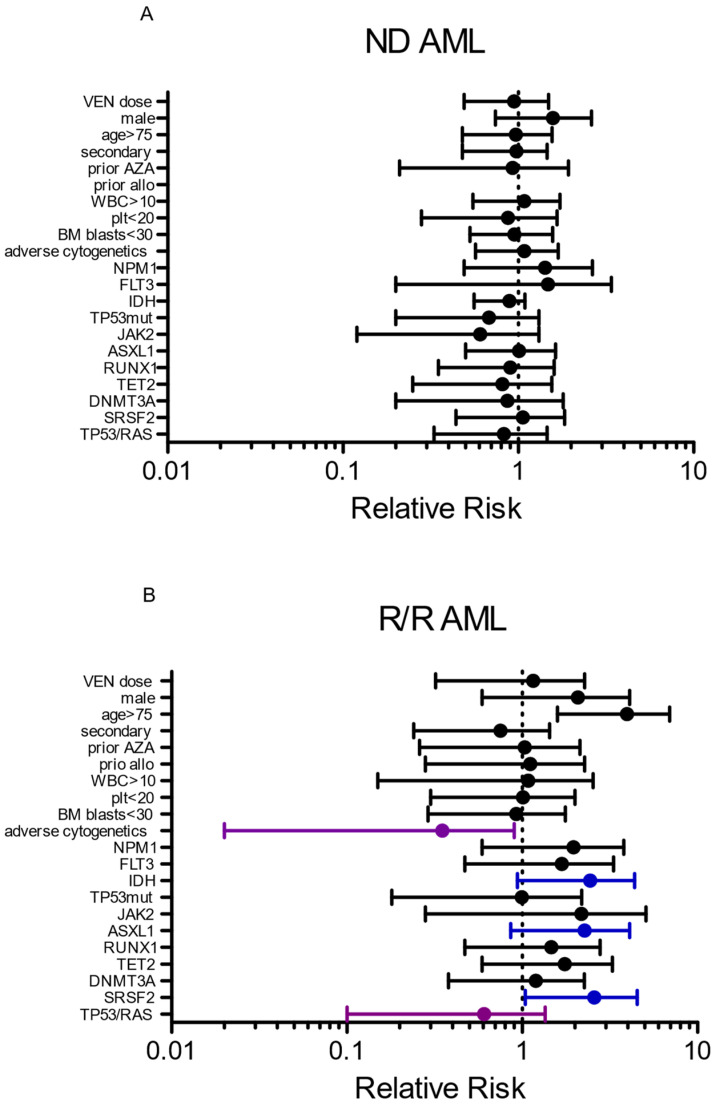
Forest plots showing clinical and molecular factors associated with response (RR > 1) or the lack of response (RR < 1) in the ND AML group (**A**) and the R/R AML group (**B**). Blue and purple lines indicate risk factors associated with response or lack of response, respectively.

**Table 1 cancers-14-02025-t001:** Patients clinical and molecular characteristics.

	Total (*n* = 77)		ND (*n* = 38)		R/R (*n* = 39)		
	*N*	%	*N*	%	*N*	%	*p* Value
Patients characteristics							
Male	45	58	21	55	24	62	0.54
Age, median (range)	72	(22–86)	73	(61–81)	69	(22–86)	<0.001
WBC, median (range)	3	(0.1–73)	4	(0.4–63)	2.9	(0.1–73)	0.153
ANC, median (range)	0.8	(0–19)	1.05	(0–19)	0.65	(0–10)	0.011
plt count, median (range)	48	(3–471)	90	(3–365)	23	(7–471)	0.221
BM blasts, median (range)	31	7–92	36	7–88	31	8–92	0.827
AML classification							
Secondary AML	47	61	28	74	19	49	0.035
AML-MRC	27	35	15	39	12	31	-
therapy-related	7	9	4	11	3	8	-
post MPN	13	17	9	24	4	10	-
Previous treatments							
Azacitidine	16	21	6	16	10	26	0.401
median cycle (range)	6	(3–20)	6	(3–13)	5	(3–20)	-
Chemotherapy	33	43	-	-	33	85	-
Median number of line	1	1–4	-	-	1	1–4	-
Allogenic transplantation	10	13	-	-	10	26	-
Cytogenetics							
Adverse cytogenetics	36	47	22	58	14	36	0.468
monosomal	24	31	13	34	11	28	0.628
complex	23	30	13	34	10	26	0.462
Genomic alteration							
*NPM1*	9	12	3	8	6	15	0.481
*FLT3*	9	12	2	5	7	18	0.154
ITD	6	8	0	0	6	15	-
TKD	3	4	2	5	1	3	-
*IDH* (*n* = 75)	18	24	8	21	10	27	0.827
*IDH1*	10	13	7	18	3	8	-
*IDH2*	8	11	1	3	7	19	-
*TP53* (*n* = 68)	15	22	8	23	7	22	1
*JAK2* (*n* = 62)	10	16	8	24	2	7	0.092
*ASXL1* (*n* = 54)	18	33	14	47	4	17	0.162
*RUNX1* (*n* = 54)	18	33	10	33	8	33	1
*TET2* (*n* = 54)	15	28	8	27	7	29	1
*DNMT3A* (*n* = 54)	14	26	6	20	8	33	0.353
*RAS* (*n* = 54)	14	26	8	27	6	25	1
*NRAS*	11	20	7	23	4	17	-
*KRAS*	3	6	1	3	2	8	-
*SFSR2* (*n* = 54)	12	22	7	23	5	21	1

**Table 2 cancers-14-02025-t002:** Multivariate analyses showing predictors for overall response rate, overall survival and event-free survival.

Overall Response Rate	Multivariate			
Variable	OR	Confidence Interval		*p* Value
		Inferior	Superior	
age	1.038	0.99	1.089	0.12
prior vidaza exposure	0.404	0.108	1.507	0.177
status (R/R versus ND)	0.403	0.131	1.238	0.113
adverse cytogenetics	0.404	0.139	1.171	0.095
Overall survival	Multivariate			
variable	HR	Confidence interval		*p* value
		Inferior	Superior	
age	1.011	0.986	1.037	0.393
prior vidaza exposure	0.885	0.416	1.886	0.752
status (R/R versus ND)	0.483	0.247	0.944	0.033
adverse cytogenetics	0.442	0.227	0.862	0.017
Event-free survival	Multivariate			
variable	HR	Confidence interval		*p* value
		Inferior	Superior	
age	0.99	0.971	1.01	0.319
prior vidaza exposure	0.756	0.389	1.472	0.411
status (R/R versus ND)	0.499	0.27	0.923	0.027
adverse cytogenetics	0.466	0.262	0.83	0.009

## Data Availability

The data presented in this study are available on request from the corresponding author.

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
