# Peer review of "Azacitidine Plus Venetoclax for the Treatment of Relapsed and Newly Diagnosed Acute Myeloid Leukemia Patients"

_cancers, 2022, doi:10.3390/cancers14082025_

Round 1

Reviewer 1 Report

The authors of the article „Azacitidine plus Venetoclax for the treatment of relapsed and First-line AML patients” provide retrospective analysis and treatment outcomes in AML patients. The objective of the study was to describe cohorts of newly diagnosed and refractory/relapsed patients and to compare clinical and molecular characteristics associated with clinical response and outcomes.

Venetoclax is widely used for the treatment of AML not only in the elderly group. Recently many articles were published, and several combinations of venetoclax with other targeted therapies become a new standard of care. Many prospective trials are ongoing with different drug combinations.

  • The authors compared outcomes of two groups- newly diagnosed (ND) AML and R/R group However they claim that the main goal of the paper is to find the predictive factors of response in the relapsed setting. If yes, why do they show the data in de novo diagnosed patients? The data of all patients can be analyzed to search for the predictors of the response and the feature “de novo/relapse” could be used as a factor in the multivariable analysis. I am not sure if from the statistical point of view, one may perform the multivariable analysis in a small group of 39 patients? Besides in the statistical methods, there is no information on how the relative risk was calculated-in fact OR should be calculated not RR. Therefore, the conclusion that adverse cytogenetic is an independent negative predictor for the response to AZA/VEN is not supported by the well-performed analysis. The same applies to other results presented in the multivariable analysis such as prior exposure to aza which in this analysis did not influence the response whereas recent data presented during ASH 2021 conference indicate that the patients who were previously treated with AZA in monotherapy  did not benefit significantly after the addition of venetoclax, and the median OS in this group was 5,8months
  • It is not clear from the paper how many patients received venetoclax shorter than 14 days . The median of venetoclax exposition was 21 days, however, some patients received only 3 days of Ven. Patients that received less than 14 days of venetoclax during the first cycle should be excluded (or at least there should be information on the number of patients). The first cycle is crucial and in other studies, authors underly the impact of time of VEN exposition on treatment outcomes.
  • Time to relapse should be added
  • Why the relapsed patients with FLT3 ITD did not receive FLTs inhibitors?
  • As the groups are very small in all OS/LFS authors should place the table with the remaining numbers at risk.
  • Why azoles were given to the patients? It is not clear if the exposition of patients treated with azoles and venetoclax at a dose 100mg is equivalent to 400mg 12 patients were FLT3+ there is no information regarding FLT3 inhibitor use
  • BM blasts in some patients were 7-8% (tablet 1) at the time of diagnosis. Was AML diagnosis based on molecular or cytogenetic changes in those cases?
  • 71% received a second course of Aza-Ven, why the treatment was stopped in the remaining 29%? Toxicity? Progression? The authors reported 12 deaths (15%) during the first 56 days of treatment.

Author Response

The authors of the article „Azacitidine plus Venetoclax for the treatment of relapsed and First-line AML patients” provide retrospective analysis and treatment outcomes in AML patients. The objective of the study was to describe cohorts of newly diagnosed and refractory/relapsed patients and to compare clinical and molecular characteristics associated with clinical response and outcomes.

Venetoclax is widely used for the treatment of AML not only in the elderly group. Recently many articles were published, and several combinations of venetoclax with other targeted therapies become a new standard of care. Many prospective trials are ongoing with different drug combinations.

We thank the reviewer for its time and expertise in reviewing our study. We have carefully considered all the reviewer’s requests for clarification below.

  • The authors compared outcomes of two groups- newly diagnosed (ND) AML and R/R group However they claim that the main goal of the paper is to find the predictive factors of response in the relapsed setting. If yes, why do they show the data in de novo diagnosed patients? The data of all patients can be analyzed to search for the predictors of the response and the feature “de novo/relapse” could be used as a factor in the multivariable analysis. I am not sure if from the statistical point of view, one may perform the multivariable analysis in a small group of 39 patients? Besides in the statistical methods, there is no information on how the relative risk was calculated-in fact OR should be calculated not RR. Therefore, the conclusion that adverse cytogenetic is an independent negative predictor for the response to AZA/VEN is not supported by the well-performed analysis. The same applies to other results presented in the multivariable analysis such as prior exposure to aza which in this analysis did not influence the response whereas recent data presented during ASH 2021 conference indicate that the patients who were previously treated with AZA in monotherapy  did not benefit significantly after the addition of venetoclax, and the median OS in this group was 5,8months

We thank the reviewer for these comments and suggestions. To address the question of predictors of response in the VEN-AZA treated patients, we performed a multivariate analysis on the whole population (n=77). We analyzed the effect of older age, disease status, adverse cytogenetics, and prior exposure to vidaza on response rate, event-free survival, and overall survival. We decided not to include molecular data in the multivariate analyses because of missing data and small sample sizes. We did a logistic regression and calculated the Odd Ratio (OR) considering the overall response rate (CR/CRi/PR/MLFS). We also did a Cox regression and calculated the Hazard Ratio (HR) for Overall Survival (OS) and Event Free Survival (EFS). As anticipated by the reviewer, disease status and adverse cytogenetics were independent risk factors associated with a trend to a lower response rate and lower survival rates in the global population. We included this analyze in the paper (section 3.5 and Table 2). In line with what the reviewer suggested, we decided not to perform a multivariate analysis on the subgroups of patients ND or R/R patients because of the small samples size (n=38 and 39, respectively). In this analyze, we chose to calculate the risk relative measuring the probability of experiencing a good response in exposed group/ probability of event in not exposed group. We completed the material & Methods section as asked by the reviewer.

Table 2 Mutivariate analyses showing predictors for overall response rate, overall survival and event-free survival

Response

Multivariate

variables

OR

Confidence interval

P value

Inferior

Superior

age

1.038

0.99

1.089

0.12

prior vidaza exposure

0.404

0.108

1.507

0.177

status (R/R versus ND)

0.403

0.131

1.238

0.113

adverse cytogenetics

0.404

0.139

1.171

0.095

Overall survival

Multivariate

variables

HR

Confidence interval

P value

Inferior

Superior

age

1,011

0,986

1,037

0,393

prior vidaza exposure

0,885

0,416

1,886

0,752

status (R/R versus ND)

0,483

0,247

0,944

0,033

adverse cytogenetics

0,442

0,227

0,862

0,017

Event-free survival

Multivariate

variables

HR

Confidence interval

P value

Inferior

Superior

age

0,99

0,971

1,01

0,319

prior vidaza exposure

0,756

0,389

1,472

0,411

status (R/R versus ND)

0,499

0,27

0,923

0,027

adverse cytogenetics

0,466

0,262

0,83

0,009

  • It is not clear from the paper how many patients received venetoclax shorter than 14 days. The median of venetoclax exposition was 21 days, however, some patients received only 3 days of Ven. Patients that received less than 14 days of venetoclax during the first cycle should be excluded (or at least there should be information on the number of patients). The first cycle is crucial and in other studies, authors underly the impact of time of VEN exposition on treatment outcomes.

We thank the reviewer for this question raising the crucial point of dose efficacy of VEN-AZA, especially during the first cycles. A recent study addressed this question (Fleishmann M et al., 2022). In our cohort, only 4 patients received less than 14 days (3 patients had to stop VEN because of rapid progression and one received only 7 days because of poor general status). We added this result in the section 3.2.

We wanted to assess the impact of VEN dosage and duration on response and survival. For this analysis, we excluded patients with severe renal impairment (n=4). We considered that the 100% of the initial VEN dose was achieved when patients received one complete cycle of 28 days of VEN 400 mg (or 100 mg if associated with azoles). Fifty-one patients received more than 50% of the target dose and 22 patients received less than 50% of the maximum dose. Interestingly, we did not observe a difference in response rate and survival between these two groups. These results were added in the section 3.3, Figure 3F, Table S3 and Fig S3.

  • Time to relapse should be added

We thank the reviewer for this suggestion. Time to relapse (=time to progression) was measured as the interval between the start of treatment and relapse after censoring death before relapse and lack of response. In our study was 203 (range 94-400) days. We added this result in the section 3.4.

  • Why the relapsed patients with FLT3 ITD did not receive FLTs inhibitors?

As noticed by the reviewer, nine patients (12%) of the patients had a FLT3 mutation (7 in the R/R group and 2 in the ND group). None of patients in the ND group received FLT3 inhibitor upfront in combination with VEN-AZA but all the R/R patients received FLT3 inhibitors (midaustorin or gilteritinib) monotherapy at relapse or in combination with chemotherapy. We apologies to the reviewer not to have provided this important information and we added it in the 3.1 section of the manuscript.

  • As the groups are very small in all OS/LFS authors should place the table with the remaining numbers at risk.

We thank the referee for this comment. We added this data on the bottom of each survival graphs of the new Figure 4.

  • Why azoles were given to the patients? It is not clear if the exposition of patients treated with azoles and venetoclax at a dose 100mg is equivalent to 400mg 12 patients were FLT3+ there is no information regarding FLT3 inhibitor use

We thank the referee for this comment. Azoles were given as antifungal prophylaxis; we added this information in section 3.2.As mentioned previously nine patients (12%) of the patients had a FLT3 mutation (7 in the R/R group and 2 in the ND group). None of patients in the ND group received FLT3 inhibitor upfront but all the R/R patients received FLT3 inhibitor (midaustorin or gilteritinib) in combination with chemotherapy or at first relapse.

  • BM blasts in some patients were 7-8% (tablet 1) at the time of diagnosis. Was AML diagnosis based on molecular or cytogenetic changes in those cases?

We thank the referee for this comment. Patients with less than 20% blasts (n=13) at the time they started VEN-AZA were patients with R/R AML (n=9) or ND patients (n=4). Regarding the R/R patients, diagnosis was based on the recurrence or the persistence of bone marrow blasts>5% after a prior line of treatment. Regarding ND AML patients, diagnosis was made on AML associated molecular or cytogenetics changes (n=8) or the presence of >20% circulating blasts (n=1).

  • 71% received a second course of Aza-Ven, why the treatment was stopped in the remaining 29%? Toxicity? Progression? The authors reported 12 deaths (15%) during the first 56 days of treatment.

We apologies if this data were not clearly indicated in the manuscript, we reformulated the sentence in the section 3.2. “Reasons for VEN-AZA discontinuation after cycle 1 for 23 patients were death (9 patients, 11.6%), progression (7 patients, 9.1%), toxicities (6 patients, 7.7%) or loss of follow up (1 patient, 1.3%). We hope that this is much clearly explained, and we thank the reviewer for requiring these precisions.

Reviewer 2 Report

Sylvain Garciaz and colleagues present a comprehensive analysis of their experience with Azacitidine and Venetoclax in newly diagnosed and in relapsed or refractory AML patients. They investigated 38 ND AML patients and 39 r/r AML patients also considering cytogenetic as well as molecular genetic risk factors.

Major comments:

  1. Table 1 is difficult to read and should be improved, e.g. by different sections.

Furthermore, absolute neutrophil counts at diagnosis should be included.

The numbers need to be checked, e.g. patients with NPM1 mutations 10 instead of 9

and some statistical considerations (ND AML vs. r/r AML) should be considered.

  1. The authors present comprehensive molecular data but focus on response analysis for single parameters only (Figure 3). In consideration that distinct mutations (e.g. PTPN11, NRAS, KRAS) can confer resistence to venetoclax, I suggest to analyse response and survival for such “cumulative subgroups”.
  2. In Figure 5B, no significant OS difference can be demonstrated while there is a clear separation of both curves. What are the reasons for early mortality in both subgroups ?
  3. The authors indicate that venetoclax was given dependent on age and co-morbidities with a range between 14 to 28 days. This has a potential impact of treatment efficacy and should by analysed in more detail. I suggest to calculate the percentage of maximum venetoclax for cycle 1 and 2 for each patient considering co-medication (e.g. 100 mg as 100% in case of posaconazole prophylaxis). Based on this, the consecutive analysis of a potential association between the “cumulative dose level” and response to Aza/VEN to initial treatment of ND AML or r/r AML will be of great value for clinicians applying this treatment regimen.
  4. Beside achievement of CR/CRi or PR haematological improvement especially the impact of Aza/VEN on transfusion frequency should by presented.
  5. Some clinically relevant non-hematological adverse events (e.g. cardiac toxicity or liver toxicity) should be included in this manuscript if easily available.

Minor comments:

  1. Figure 2 and Figure 4 should be combined in one figure.

In Figure 2 (A, B and C), N needs to replaced by %.

  1. Figure 5 and Figure 6 should be combined in one figure.
  2. The authors should consider to include the following papers (Zhao P et al. and Fleischmann M et al. – each published in 2022) on HMA/VEN treatment in their discussion:

https://pubmed.ncbi.nlm.nih.gov/34568973/

https://pubmed.ncbi.nlm.nih.gov/35099591/

The manuscript is well written and has a potential impact on a better understanding of this important treatment regimen if the above aspects can be addressed.

Author Response

Sylvain Garciaz and colleagues present a comprehensive analysis of their experience with Azacitidine and Venetoclax in newly diagnosed and in relapsed or refractory AML patients. They investigated 38 ND AML patients and 39 r/r AML patients also considering cytogenetic as well as molecular genetic risk factors.

We thank the reviewer for its time and expertise in reviewing our study. We have carefully considered all the reviewer’s requests for clarification below.

Major comments:

  1. Table 1 is difficult to read and should be improved, e.g. by different sections.

Furthermore, absolute neutrophil counts at diagnosis should be included.

The numbers need to be checked, e.g. patients with NPM1 mutations 10 instead of 9and some statistical considerations (ND AML vs. r/r AML) should be considered.

We apologies if Table 1 was not easy to read. We added the requested information (ANC, number of NPM1 mutation and statistical comparisons between ND and RR groups). We split the table into different sections. We hope this improved the comprehension of Table 1.

The authors present comprehensive molecular data but focus on response analysis for single parameters only (Figure 3). In consideration that distinct mutations (e.g. PTPN11, NRAS, KRAS) can confer resistence to venetoclax, I suggest to analyse response and survival for such “cumulative subgroups”.

We thank the reviewer for this very nice suggestion. Based on the cohort of patient with available NGS data (n=54), NRAS and KRAS mutations represent 14 patients, PTPN11 has been found mutated in 4 patients, all of them harboring a concomitant NRAS mutation. We pooled the 14 RAS mutated patients with the 10 mutated TP53mutated patients to obtain a group of 22 patients (2 patients had both a RAS and a TP53 mutation). This subgroup with poor molecular risk had a lower response rate (Supplementary Table 4, new Figure 3E) and a lower survival (new Figure 4E-F). Given the small patient size with molecular data available, we did not perform additional analysis on this cohort. We are planning to collect data on a bigger cohort to perform further analyses.

2. In Figure 5B, no significant OS difference can be demonstrated while there is a clear separation of both curves. What are the reasons for early mortality in both subgroups ?

We thank the referee for this comment. We registered 13 deaths in the ND cohort within the first 6 months after treatment initiation. Two patients died from non-relapse mortality (sepsis). The other patients relapsed rapidly after VEN-AZA initiation and died because of the progression of the disease. Most of these patients had poor-risk cytogenetics status (9/13) or TP53/RAS mutations (7/13). We had added these data in the 3.4 section.

3. The authors indicate that venetoclax was given dependent on age and co-morbidities with a range between 14 to 28 days. This has a potential impact of treatment efficacy and should by analysed in more detail. I suggest to calculate the percentage of maximum venetoclax for cycle 1 and 2 for each patient considering co-medication (e.g. 100 mg as 100% in case of posaconazole prophylaxis). Based on this, the consecutive analysis of a potential association between the “cumulative dose level” and response to Aza/VEN to initial treatment of ND AML or r/r AML will be of great value for clinicians applying this treatment regimen.

We thank the reviewer for this question raising the crucial point of dose efficacy of VEN-AZA, especially during the first cycles. A recent study addressed this question (Fleishmann M et al., 2022). In our cohort, only 4 patients received less than 14 days (3 patients had to stop VEN because of rapid progression and one received only 7 days because of poor general status). We added this result in the section 3.2.

We wanted to assess the impact of VEN dosage and duration on response and survival. For this analyze, we excluded patients with severe renal impairment (n=4). We considered that the 100% of the initial VEN dose was achieved when patients received one complete cycle of 28 days of VEN 400 mg (or 100 mg if associated with azoles). Interestingly, 51 patients received more than 50% of the target dose and 22 patients received less than 50% of the dose. We did not observe a difference in response rate and survival between these two groups. These results were added in the section 3.3, Figure 3F, Table S3 and Fig S3.

4. Beside achievement of CR/CRi or PR haematological improvement especially the impact of Aza/VEN on transfusion frequency should by presented.

The Institut Paoli-Calmettes is a tertiary referral hospital, as a consequence, a large part of the patients is referred for AML treatments. We don’t easily get the transfusion baseline from the prior 3 months for most of them. Therefore, we cannot use the Gale criteria, with dependence defined as ≥ 2 units/month for ≥ 3 months (Gale RP, et al., What are RBC-transfusion-dependence and -independence? Leuk Res. 2011 Jan;35(1):8-11). Therefore, transfusion frequency after VEN-AZA treatment was not easily interpretable. We are planning to collect this data on a cohort of patients with baseline transfusion frequency data available.

5.Some clinically relevant non-hematological adverse events (e.g. cardiac toxicity or liver toxicity) should be included in this manuscript if easily available.

Another work is currently ongoing in a larger cohort of AML patients treated with VEN, specifically assessing toxicity and tolerance all along VEN-AZA treatment. This analyze will consider hematological and non-hematological adverse events depending on the dose of VEN and the use of azoles. We hope the reviewer will understand we decided not to provide incomplete data in this current report.

Minor comments:

  1. Figure 2 and Figure 4 should be combined in one figure.

We mixed Figure 2 and 4 together and designed a new Figure 3. To keep consistency between the displayed bar graphs, we decided to show the evaluation of response for the whole cohort of patient in the new Figure 3 and to remove the figure showing response rates among cytogenetic groups. We are showing these data in the forest plots (new Figure 5)

In Figure 2 (A, B and C), N needs to replaced by %.

We thank the referee for point out this mistake. We replaced N by %, accordingly.

2.Figure 5 and Figure 6 should be combined in one figure.

We drew a new Figure 4 combining the survival curves from old Figures 5 and 6.

3. The authors should consider to include the following papers (Zhao P et al. and Fleischmann M et al. – each published in 2022) on HMA/VEN treatment in their discussion:

https://pubmed.ncbi.nlm.nih.gov/34568973/

https://pubmed.ncbi.nlm.nih.gov/35099591/

We thank the referee for these references. We added these two references in the discussion part (lines 282 and 294). 

The manuscript is well written and has a potential impact on a better understanding of this important treatment regimen if the above aspects can be addressed.

We thank the referee for its time and expertise in reviewing our work. We are delighted that the referee feels the overall message is interesting.

Reviewer 3 Report

The manuscript (ID:  cancers-1651045) entitled “Azacitidine plus Venetoclax for the Treatment of Relapsed and First-line AML Patients” by Dr. Garciaz and colleagues reports on a comparison of the outcome of n=39 relapsed and/or refractory AML patients and n=38 concomitant newly diagnosed AML patients, for a total of n=77 patients, under combinatorial treatment with Venetoclax and azacitidine. Main results indicate that response rates were lower in relapsed and/or refractory AML group, while adverse cytogenetics was associated with treatment failure only in the relapsed and/or refractory group. ASXL1, IDH and SFSR2 mutations were associated with a trend in a higher response rate in the relapsed and/or refractory group. Median survival of responding patient was not different between relapsed and/or refractory and newly diagnosed AML patients (9 months). The work is interesting, concise, and well written. The experimental design is well performed. It will increase our knowledge on AML therapy with BCL2 inhibitors and hypomethylating agents.

In my opinion, the ms can be accepted with a minor revision. I have several comments for improving the manuscript:

Comments

  1. I suggest replacing “AML” with “acute myeloid Leukemia” in the title for a better reading
  2. A conclusive sentence should be included in the abstract
  3. If patients were from a clinical trial, this information should be included in the method section
  4. Supporting references should be included in the “Patient samples molecular characterization” and “statistical analysis”sections
  5. A figure summarizing the drug schedule would be helpful for the reader as it could increase the quality of the work

Minor observations

Line 24 AML should be Acute myeloid leukemia (AML) when mentioned for the first time

Lines 49-53 Combination therapies with venetoclax and azacitidine, including mechanisms of resistance, have also extensively been described/discussed here (doi: 10.1038/s41375-021-01218-0.) For completeness, this reference should be included

Lines 57-69 The sentence is lacking in supporting references. This reference should be included (DOI: 10.1016/S1470-2045(18)30010-X)

Line 94 “as previously described” ref?

Line 95 ddPCR should be droplet-digital PCR (ddPCR) when mentioned for the first time

Author Response

The manuscript (ID:  cancers-1651045) entitled “Azacitidine plus Venetoclax for the Treatment of Relapsed and First-line AML Patients” by Dr. Garciaz and colleagues reports on a comparison of the outcome of n=39 relapsed and/or refractory AML patients and n=38 concomitant newly diagnosed AML patients, for a total of n=77 patients, under combinatorial treatment with Venetoclax and azacitidine. Main results indicate that response rates were lower in relapsed and/or refractory AML group, while adverse cytogenetics was associated with treatment failure only in the relapsed and/or refractory group. ASXL1, IDH and SFSR2 mutations were associated with a trend in a higher response rate in the relapsed and/or refractory group. Median survival of responding patient was not different between relapsed and/or refractory and newly diagnosed AML patients (9 months). The work is interesting, concise, and well written. The experimental design is well performed. It will increase our knowledge on AML therapy with BCL2 inhibitors and hypomethylating agents.

In my opinion, the ms can be accepted with a minor revision. I have several comments for improving the manuscript:

We are delighted that the referee feels the overall message is interesting. We thank the reviewer for its time and expertise in reviewing our study. We have carefully considered all the reviewer’s requests for clarification below.

Comments

  1. I suggest replacing “AML” with “acute myeloid Leukemia” in the title for a better reading

We thank the reviewer for this suggestion. We made the change.

  1. A conclusive sentence should be included in the abstract

We are totally agreeing with the reviewer. We added a conclusive sentence.

  1. If patients were from a clinical trial, this information should be included in the method section

Patients were not included in a clinical trial. We apologies if this was not clearly indicated.

  1. Supporting references should be included in the “Patient samples molecular characterization” and “statistical analysis”sections

We thank the reviewer for this comment. We added a reference in regard with the molecular characterization. We added details in regards with commercial kits used for molecular analyses and added a supplementary Table 1 including the myeloid panel of genes used for NGS. We also completed the statistical analysis section with selected references.

  1. A figure summarizing the drug schedule would be helpful for the reader as it could increase the quality of the work

We thank the referee for this suggestion. We made a new Figure 1, summarizing the drug schedule.

Minor observations

Line 24 AML should be Acute myeloid leukemia (AML) when mentioned for the first time

We thank the referee for point out this error – it has been corrected

Lines 49-53 Combination therapies with venetoclax and azacitidine, including mechanisms of resistance, have also extensively been described/discussed here (doi: 10.1038/s41375-021-01218-0.) For completeness, this reference should be included

We added this interesting review in the introduction and the discussion parts of the manuscript.

Lines 57-69 The sentence is lacking in supporting references. This reference should be included (DOI: 10.1016/S1470-2045(18)30010-X)

We added this major reference, we thank the referee.

Line 94 “as previously described” ref?

We added a reference from a previous work from our group.

Line 95 ddPCR should be droplet-digital PCR (ddPCR) when mentioned for the first time

We thank the referee for point out this error – it has been corrected

Round 2

Reviewer 1 Report

Thank You for addressing all questions raised during the reviewing process The paper can be accepted in the current form